# Effect of Group Antenatal Care on Breastfeeding Knowledge and Practices Among Pregnant Women in Ghana: Findings from a Cluster-Randomized Controlled Trial

**DOI:** 10.3390/ijerph21121587

**Published:** 2024-11-28

**Authors:** Theresa Norpeli Lanyo, John Williams, Bidisha Ghosh, Veronica E. A. Apetorgbor, Vida A. Kukula, Ruth Zielinski, Elizabeth Awini, Cheryl Moyer, Jody R. Lori

**Affiliations:** 1Department of Health Behavior and Biological Sciences, University of Michigan School of Nursing, 400 North Ingalls, Ann Arbor, MI 48109-2003, USA; 2Dodowa Research Center, Ghana Health Service, Dodowa AZ820, Greater Accra, Ghana; 3Department of Learning Health Sciences, University of Michigan Medical School, 2054 1111 E. Catherine St., Ann Arbor, MI 48109-2054, USA

**Keywords:** breastfeeding, group antenatal care, maternal healthcare, child survival

## Abstract

The World Health Organization (WHO) recommends exclusive breastfeeding for the first six months after birth to ensure child health and survival. Antenatal care provides an opportunity to educate pregnant women on optimal breastfeeding practices. A cluster-randomized control trial in Ghana examined the impact of group antenatal care on breastfeeding knowledge and practice. The study enrolled 1761 pregnant women from 14 health facilities in Ghana. The intervention group (*n* = 877) received eight group sessions, while the control group (*n* = 884) received individual, routine care. Data were collected at baseline and post intervention. Pearson’s chi-square test was performed to examine categorical data, while odds ratios were calculated using separate logistic regression models to examine differences between the intervention and control groups over time. Women enrolled in group antenatal care had higher odds of following WHO recommendations to exclusively breastfeed for the first six months (odds ratio [OR]: 3.6, 95% confidence interval [95% CI]: 2.1, 6.3) and waiting to introduce solid food until six months of age (OR: 3.1, 95% CI: 1.5, 6.9). Our results found that women who participated in group antenatal care were more likely to follow the recommendations for exclusive breastfeeding developed by the WHO.

## 1. Background

Antenatal care (ANC) is one of the most utilized preventive maternal healthcare services with an impact on lowering the rates of maternal mortality and morbidity (MMR), along with the reduction in stillbirths and perinatal deaths [1]. The World Health Organization (WHO) outlined a comprehensive global guideline for ANC practice covering various areas, including breastfeeding as a preventive measure for both mother and baby [1]. Proper nutrition is essential for infants to achieve optimal growth and development, and breastfeeding exclusively is considered one of the most cost-effective and convenient ways to ensure child health and survival [2]. WHO and the United Nations International Children’s Emergency Fund (UNICEF) recommend that breastfeeding begin within one hour of birth. Infants should be exclusively breastfed for the first six months (with no other liquids or food, including water) and should additionally continue breastfeeding until two years of age or beyond while also introducing appropriate and safe solid (complementary) foods [3]. According to WHO, breastmilk is safe and clean, requires no preparation, even in environments with poor sanitation and unsafe drinking water, and contains all the essential nutrients and antibodies to protect against childhood illnesses [2]. This makes breastmilk the ideal food for infants for the first six months up to the second year of life [4]. Optimal breastfeeding is the best source of nutrition for the newborn and the cornerstone for establishing healthy growth and development for children [5,6]. It has beneficial effects on a child’s cognitive development, protection against obesity, respiratory tract illnesses, bronchial asthma, type 2 diabetes, cholesterol regulation, and reduction in diarrhea problems [2,7,8,9]. Infants who are not breastfed are 6 to 10 times more likely to die in the early months than those who are exclusively breastfed [10]. Breastfeeding is a potential measure to annually save the lives of 2.7 million from stunting, wasting, overweight, and underweight [10,11]. Undernutrition contributes to 45% of all infant deaths globally [10,11]. The majority of these deaths are accounted for by low- and middle-income countries (LMICs) in the sub-Saharan Africa (SSA) region and southern Asia [2,11], where exclusive breastfeeding (EBF) is practiced sub-optimally, with less than half of infants under 6 months exclusively breastfeeding [2]. Additionally, access to clean water, adequate sanitation, and essential health and social services are often limited in SSA [4,12].

### 1.1. Antenatal Care and Breastfeeding

Addressing the health and practical benefits and importance of exclusive breastfeeding remains essential for healthcare research and practice. There is mounting evidence that ANC interventions, alone or in combination with intrapartum and postpartum support, can increase breastfeeding initiation, duration, and exclusivity through the provision of adequate information [13,14,15]. Studies have shown that only a fraction of clients receive adequate breastfeeding counseling during pregnancy [16]. Additional studies have identified inappropriate communication [17], lack of knowledge [4,12], outdated health facilities and cultural practices [12], and healthcare provider (HCP) shortage (lack of skilled birth attendants) [18,19] as barriers leading to little time spent on client counseling [20,21], and oversights in providing the required support to mothers for improving breastfeeding practices [22,23].

### 1.2. Group Antenatal Care and Breastfeeding

Breastfeeding can be challenging, particularly in the first moments after birth. However, having the right policies, programs, and people in place provides a strong support network for mothers. The group antenatal care (GANC) intervention was based on the Health Literacy Skills Framework [24]. An initial feasibility study [25] was conducted using this framework to assess and tailor the implementation of GANC in Ghana. Studies have shown that social and professional support and follow-up improve breastfeeding practices among mothers [26,27,28]. The holistic and patient-centered nature of group antenatal care (GANC) makes it possible to provide comprehensive ANC that can culturally tailor counseling and support toward breastfeeding. Implementing GANC in LMICs is relatively new, with only a handful of studies examining whether GANC affects breastfeeding. This study examines the impact of GANC on breastfeeding knowledge and practices among mothers in Ghana.

## 2. Method

This study was a cluster-randomized control trial conducted from July 2019 to May 2022 in the Eastern Region of Ghana using 14 matched-pair health facilities in four [4] districts. These facilities provide focused antenatal services to people within the catchment area. The trial was registered with ClinicalTrials.gov on 25 July 2019 (RCT: NCT04033003). The protocol for this trial can be found elsewhere [29].

The study setting includes four districts within the Eastern Region of Ghana: Akwapim North, Yilo Krobo, Nswam-Adoagyiri, and Lower Manya Krobo. Intervention and control sites covered distinct catchment communities to prevent the overlap of study participants. The study used purposive sampling to identify pregnant women who enrolled in ANC at the randomized facilities within the selected region of Ghana. Randomization was achieved using a matched-pair method based on the number of deliveries and average gestation at the time of ANC enrollment at each facility. The nbpMatching package from R software (version 1.5.0; R Foundation for Statistical Computing) was used to complete the matching and randomization process [30]. A sample size of about 100 women per facility was calculated at 80% power with an intraclass correlation coefficient of 0.01 [31]. To account for attrition, 120 women per facility were recruited [29]. Pregnant women who attended any of the 14 selected facilities with a gestation of less than 20 weeks, over 15 years old, spoke any of the following languages: English, Ewe, Dangme, Ga, or Akan, and consented to participate in the study were offered the opportunity to participate in the study. A trained research assistant (RA) then provided a detailed explanation of the study and obtained informed consent. Those with high-risk pregnancies requiring specialist care were excluded. Figure 1 shows the flow of participants through the different stages of the study.

### 2.1. Group Antenatal Care Intervention

The group antenatal care intervention occurred in seven randomized health facilities. A comprehensive individual assessment was conducted at the initial visit, followed by the eight group sessions. The sessions follow the eight contact visits recommended by WHO [1], which is currently practiced in Ghana. Each session lasted approximately 60–90 min. A midwife led the sessions in groups of 12–14 pregnant women. The sessions comprised evidence-based discussions and participatory learning activities tailored to pregnancy, delivery, and postpartum and newborn health, including breastfeeding. All discussions were guided by a validated Facilitator’s Guide for Antenatal Care [25]. Women enrolled in the control groups received individual antenatal care (IANC) per the standard of care for focused antenatal care.

### 2.2. Data Collection

Research assistants used password-encrypted tablets to collect data from pregnant women at various time points: Time 0: baseline (immediately following the consent process); Time 1: 34 weeks gestation—3 weeks post-delivery; Time 2: 6–12 weeks postpartum; Time 3: 5 to 8 months postpartum; Time 4: 11 to 14 months postpartum. At Times 0 and 1, questions on breastfeeding knowledge and intention were asked.

### 2.3. Data Analysis

Participants’ baseline demographic data were summarized using descriptive statistics and analyzed using bivariate statistical tests to compare the baseline characteristics of the participants between the GANC intervention and IANC control groups. Two sample *t*-tests were used to compare age and wealth index, while the Mann–Whitney Wilcoxon test was used to compare the number of previous pregnancies. Chi-squared tests were used to compare categorical data. Logistic regression models adjusted for clustering to test the differences in participants’ knowledge about the appropriate time to start breastfeeding a baby, the age at which to give additional fluids other than breastmilk, and the introduction of solid food between the two groups over time. The potential covariates were the study arms (GANC vs. IANC) and time (T0 vs. T1). Education and parity were also included to control for potential confounding.

## 3. Results

Table 1, the demographic table, presents the baseline characteristics of the participants. The majority of the participants (56%) were in the 25–34 age group. In terms of education, 10% of the participants and 16% of their partners reported tertiary education. Ninety-three (93%) of the participants were Christians, and 20% of the participants were experiencing their first pregnancy. The demographics were evenly distributed between the two study arms.

Table 2 compares the percentage increase in mothers’ knowledge of the appropriate age at which to introduce fluids other than breast milk to infants between the intervention group (GANC) and the control group (IANC). At baseline, the estimated percentage of women choosing 6 months or older to introduce fluids other than breastmilk was similar for the two groups: 81% of the women in IANC and 82.6% in GANC. At Time 1, while this percentage increased for both groups, the percentage rose more sharply in the GANC than the IANC (*p* < 0.0001). The estimated percentage of participants reporting the ideal time to introduce fluids other than breastmilk at or after 6 months of age increased from 81% to 89% in the IANC and 82% to 97% in the GANC.

A similar increase was also seen for the age at which mothers began feeding solid food to babies. At baseline, the estimated percentage of women choosing 6 months or older as the appropriate age to start giving babies solid food was about 90% for both groups. At T1, while the percentage rose for both groups, the increase was significantly greater in the GANC, where 98% of the women were estimated to choose 6 months or older, compared to 95% in the IANC (*p* < 0.0001).

Table 3, the model summary, presents the results of the study with three different outcomes, showing the odds ratios, *p*-values, and 95% confidence intervals for various factors.

### 3.1. Outcome 1: Knowledge on Initiation of Breastfeeding

Overall, no significant differences were seen between the two groups (OR = 1.0, 95% CI = 0.8, 1.3). There is an overall time effect where the initiation of breastfeeding was significantly lower at T1 compared to T0 (OR: 0.75, 95%: 0.7, 0.9), suggesting that women in GANC have the highest likelihood of initiating breastfeeding early. Tertiary-educated women had the highest probability of initiating breastfeeding within 30 min of delivery compared to less-educated women. The odds of early breastfeeding initiation are significantly lower for primary education (OR: 0.55, 95% CI: 0.37, 0.81), junior high (OR: 0.59, 95% CI: 0.42, 0.84), and secondary education (OR: 0.58, 95% CI: 0.4, 0.83).

### 3.2. Outcome 2: Best Time to Start Giving Fluids Other than Breastmilk

Mothers enrolled in GANC had greater odds of reporting exclusive breastfeeding as compared to IANC over time (*p*-value ≤ 0.0001). They reported the introduction of fluids other than breastmilk at 6 months or later (OR: 3.6, 95% CI: 2.1, 6.3). The results also found a significant effect due to parity and education. First-time mothers were more likely to introduce fluids before the recommended six months (OR: 0.52, 95% CI: 0.4, 0.7). In terms of education, women reporting a tertiary education had the highest likelihood of waiting to give baby fluids until 6 months or later compared to those with lower education such as primary (OR: 0.32, 95% CI: 0.2, 0.6), junior high (OR: 0.35, 95% CI: 0.2, 0.6), and secondary education (OR: 0.47, 95% I: 0.3, 0.8).

### 3.3. Outcome 3: Best Time to Start Giving Solid Food to Babies

Over time, mothers enrolled in GANC had greater odds of giving solids to babies at 6 months or later compared to IANC (*p*-value = 0.003). In addition, the odds of opting to introduce solids at or after 6 months are lower among first-time mothers (OR: 0.69, 95% CI: 0.4, 0.9). Also, women with tertiary education had higher odds of waiting until at least 6 months of age to introduce solid foods than women with less education: primary (OR: 0.14, 95% CI: 0.1, 0.4); junior high (OR: 0.18, 95% CI: 0.1, 0.5); secondary (OR: 0.3, 95% CI: 0.1, 0.8).

## 4. Discussion

The current study examined the impact of GANC on breastfeeding knowledge and practices among mothers in Ghana. Group antenatal care is an innovative approach to care with reports of higher patient and provider satisfaction rates and lower maternal and infant mortality and morbidity rates [1,32,33,34,35]. Mothers’ knowledge and practices play key roles in the process of breastfeeding [36]. Breastfeeding is a critical practice that is part of a complex intervention for improving maternal and infant health but lacks evidence regarding the effects of group care interventions [37]. Behavioral change interventions such as GANC have shown to be effective in increasing breastfeeding knowledge, leading to better practice behaviors [38]. Breastfeeding interventions delivered in combination with health services and group contexts double exclusive breastfeeding practice [39].

Our study results reveal that GANC improved optimal timing for introducing fluids and complementary foods. This indicates that apart from the statistical significance, this result is a clinically important finding that aligns with the WHO recommendation of exclusively breastfeeding for the first six months without other liquids or food, including water [2]. Compelling data indicates that breastfeeding protects against pneumonia and diarrhea, the primary causes of infant mortality [6,40], leading to approximately 50% reduction in diarrheal episodes and 33% of respiratory illnesses [4].

Studies have shown that women receiving individualized antenatal care have poor breastfeeding knowledge and practices compared to GANC [20,21]. Factors contributing to these issues include inadequate breastfeeding counseling during pregnancy [16], ineffective communication and support [17], outdated health facilities and cultural practices [12], and healthcare provider (HCP) shortage (lack of skilled birth attendants) usually leading to little time spent on client counseling [18,19]. Breastfeeding education and counseling through GANC have been associated with increased breastfeeding knowledge [41,42]. The support provided and garnered from GANC increases rates and improves breastfeeding practices among mothers [43]. These findings emphasize that knowledge and support from GANC increase: (1) the mother’s psychological well-being, self-confidence, motivation, intention, competence, and autonomy to breastfeed exclusively [42,44,45,46]; (2) timely initiation and exclusive breastfeeding [47,48,49]; (3) higher breastfeeding rates [50,51]; and (4) breastfeeding sustenance up to two years [52]. Improving breastfeeding knowledge not only helps to achieve better infant health [41] but also leads to cost savings (reduction in health expenditure) due to fewer infections and diseases caused by poor breastfeeding practices [53] and reduced costs in purchasing baby foods [54,55].

While our study identified positive relationships between higher levels of education and multiparity to better breastfeeding knowledge, other studies have reported that mothers have several reasons for engaging in various breastfeeding behaviors: facility delivery [56], good health status [5], vaginal delivery [26,56], partner knowledge [56], and employment status [5,26,28,56,57] reflect other reasons that facilitate exclusive breastfeeding among mothers. Some major barriers to exclusive breastfeeding, such as poor knowledge (mostly related to myths and misconceptions) [27,57], lack or improper counseling by healthcare providers [21], and lack of a partner, family, or professional support [27], were comprehensively addressed by the broad scope and impact of GANC [42,44,46,47,48,49]. This underscores the importance of using effective intervention service delivery techniques, such as GANC, to intensify counseling and support and address the diversity and complexities around maternal breastfeeding behavior for improving child health and survival.

### Strengths and Limitations

Participants were not blinded to the study. However, facilities within each matched pair randomized to either intervention or control were far apart, preventing any possibility of contamination. Participant selection was purposive at the study sites; however, randomization was implemented at the facility level to address potential issues related to generalization. Participant follow-up for this study ended at fourteen months. Additional data will be needed to assess the long-term adherence to continued breastfeeding at two years and its effects on maternal and newborn health outcomes.

The cluster-randomized design has many strengths, including evaluating effectiveness under conditions of actual use and the possibility of generalizing results to clinical practices similar to those in this trial. This design maintains the rigor and internal validity of an RCT while enhancing external validity through essential methodological features.

## 5. Conclusions

Group antenatal care has the potential to be a significant intervention in promoting better breastfeeding practices. Tackling maternal and infant health issues is complex for low-resourced areas. The provision of GANC is one strategy for managing the complexities around maternal healthcare and improving the health and well-being of mothers and their children. Future research should continue to explore the long-term effects of GANC on breastfeeding practices and maternal–infant health, as well as identify the most effective components of the program to further enhance its impact. Findings from our study can support policymakers and stakeholders in the SSA region to consider the implementation of GANC to improve maternal health and improve practices such as breastfeeding.

## Figures and Tables

**Figure 1 ijerph-21-01587-f001:**
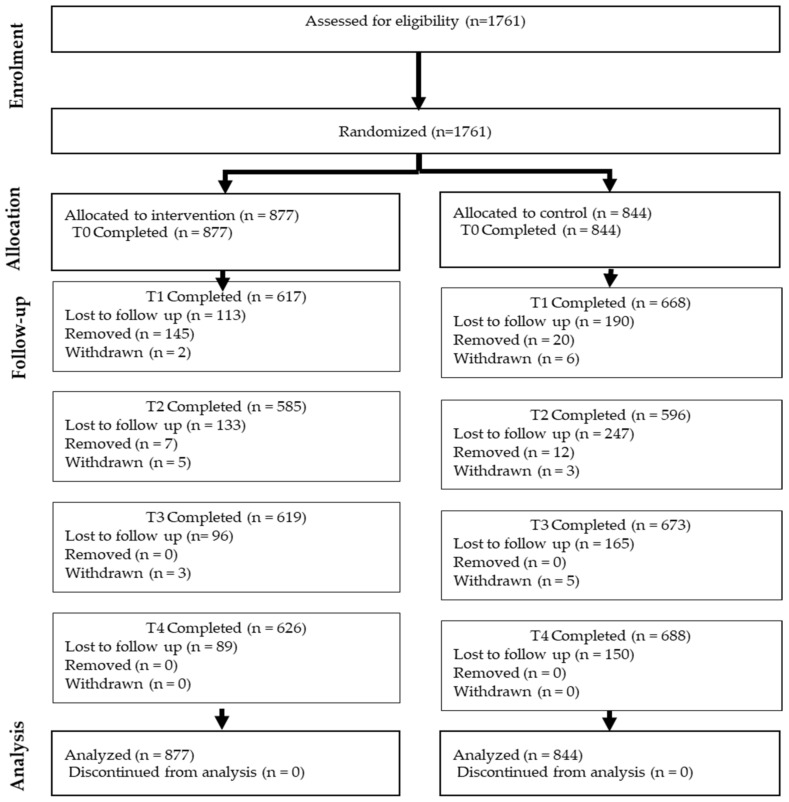
Flow diagram of the progress through the phases of a parallel randomized trial of two groups (Participants lost to follow-up at T1, T2, or T3 remain eligible to complete surveys at subsequent timepoints.).

**Table 1 ijerph-21-01587-t001:** Demographic Table.

Categorical Variables	Total (*n* = 1761)	Control (*n* = 884)	Intervention (*n* = 877)	*p*-Value
Age				
<25	501 (28%)	266 (53%)	235 (47%)	0.19
25–34	987 (56%)	477 (48%)	510 (52%)	
≥35	273 (16%)	141 (52%)	132 (48%)	
Educational Level				
Primary	246 (14%)	120 (49%)	126 (51%)	0.69
≥Junior Secondary/Junior High	829 (49%)	429 (52%)	400 (48%)	
Secondary	459 (27%)	223 (49%)	236 (51%)	
Tertiary	164 (10%)	83 (51%)	81 (49%)	
Religion				
Christianity	1646 (93%)	835 (50.4%)	811 (49.6%)	0.12
Muslim	97 (6%)	39 (39%)	58 (61%)	
Other	18 (1%)	10 (56%)	8 (44%)	
First Pregnancy				
No	1412 (80%)	703 (50%)	709 (50%)	0.49
Yes	349 (20%)	181 (52%)	168 (48%)	
Location of Delivery				
Health Facility	1711 (97%)	853 (50%)	858 (50%)	0.09
Other	50 (3%)	31 (62%)	19 (38%)	
Continuous Variables Mean (SD)				
Maternal Age	28.2 (5.8)	28.1 (6)	28.3 (5.6)	0.50
Wealth Index	6.8 (2.4)	6.9 (2.4)	6.9 (2.3)	0.62
Number of Previous Pregnancies	3.5 (1.4)	3.5 (1.4)	3.5 (1.5)	0.71
Categorical variables were compared using the chi-square test				
Maternal age and wealth index tested using the two-sample *t*-test				
Number of previous pregnancies was tested using the Mann–Whitney Wilcoxon test (non-parametric)				

**Table 2 ijerph-21-01587-t002:** Percentage increase in giving fluids and solid food at six months or more by group.

		Time = T0 *n* (%)			Time = T1 *n* (%)			
Item	Study Arm	*n*	<6 Months	≥6 Months	*n*	<6 Months	≥6 Months	*p*-Value
At what age do you think it is best to start giving your baby fluids other than breastmilk?	IANC	824	155 (18.8)	669 (81.2)	657	75 (11.4)	582 (88.6)	*p* < 0.0001
	GANC	809	144 (17.8)	665 (82.2)	613	20 (3.2)	593 (96.7)	
At what age do you think it is best to start giving your baby solid food?	IANC	834	85 (10.2)	749 (89.8)	657	34 (5.2)	623 (94.8)	*p* < 0.0001
	GANC	809	76 (9.4)	733 (90.6)	610	10 (1.6)	600 (98.4)	

**Table 3 ijerph-21-01587-t003:** Model Summary.

**Outcome 1. Giving Breastmilk at <30 Min**		**Odds Ratio**	***p*-Value**	**95% Confidence Interval**
Study Arm	IANC = 757	(reference)	0.964	
	GANC = 737	1.01		0.8, 1.3
Time	0	(reference)	0.014	
	1	0.75		0.7, 0.9
Interaction: Time and Group	(reference)		0.7318	
	Time = 1, Group = GANC	0.95		0.7, 1.3
First Pregnancy	No	(reference)	0.5531	
	Yes	1.08		0.8, 1.4
Education	Tertiary	(reference)		
	Primary	0.55	0.003	0.37, 0.81
	Junior High	0.59	0.003	0.42, 0.84
	Secondary	0.58	0.003	0.4, 0.83
**Outcome 2. Age at which you think it is best to start giving fluids to baby other than breastmilk**		**Odds Ratio**	***p*-Value**	**95% Confidence Interval**
Study Arm	IANC = 824	(reference)	0.534	
	GANC = 809	1.09		0.9, 1.4
Time	0	(reference)	<0.0001	
	1	1.89		1.5, 2.5
Interaction: Time and Group	(reference)		<0.0001	
	Time = 1, Group = GANC	3.6		2.1, 6.3
First Pregnancy	No	(reference)	<0.0001	
	Yes	0.52		0.4, 0.7
Education	Tertiary	(reference)		
	Primary	0.32	<0.0001	0.2, 0.6
	Junior High	0.35	<0.0001	0.2, 0.6
	Secondary	0.47	0.012	0.3, 0.8
**Outcome 3. The best time to start giving solid food to babies**		**Odds Ratio**	***p*-Value**	**95% Confidence Interval**
Study Arm	IANC = 834	(reference)	0.596	
	GANC = 809	1.09		0.8, 1.5
Time	0	(reference)	<0.0001	
	1	2.1		1.4, 3.1
Interaction: Time and Group	(reference)		0.003	
	Time = 1, Group = GANC	3.17		1.5, 6.9
First Pregnancy	No	(reference)	0.011	
	Yes	0.69		0.4, 0.9
Education	Tertiary	(reference)		
	Primary	0.14	<0.0001	0.1, 0.4
	Junior High	0.18	<0.0001	0.1, 0.5
	Secondary	0.3	0.012	0.1, 0.8

## Data Availability

Data are available in the Deep Blue repository of the University of Michigan at https://deepblue.lib.umich.edu/data (accessed on 27 November 2024).

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
