# Peer review of "Effect of Group Antenatal Care on Breastfeeding Knowledge and Practices Among Pregnant Women in Ghana: Findings from a Cluster-Randomized Controlled Trial"

_ijerph, 2024, doi:10.3390/ijerph21121587_

Round 1

Reviewer 1 Report

Comments and Suggestions for Authors

In Section 1.2, further characterize the content of group antenatal care (GANC) in relation to implementation in Ghana. How is it adapted in content in relation to the specific conditions of this country?

What is the time factor for implementing antenatal care overall in Ghana?

In section 2.1, add breastfeeding to the content of antenatal care.
Participation in the research (group antenatal care and individual antenatal care) was free in both groups?

Missing from the discussion is a rationale for the importance of group antenatal care over individual group care.
It would be useful to add the results of studies that focus on breastfeeding outcomes after individual group care and compare them with the outcomes given.

Reviewer 2 Report

Comments and Suggestions for Authors

Effect of Group Antenatal Care on Breastfeeding Knowledge and Practices Among Pregnant Women in Ghana: Findings from A Cluster Randomized Controlled Trial

This is a well-written, quality manuscript that deserves consideration for publication.

My detailed observations and suggestions are as follows:

1. Title - justified based on the study 

2. Authors - The first author is a PhD student, so the department and the institution need to be mentioned.

3. Abstract - described adequately.

4. Background - Mentioned the importance and beneficial effects of breastfeeding, WHO recommendations for breastfeeding, antenatal care, group antenatal care, and the study's objective.

5. Methods: This study used a cluster randomized controlled method, a standard and acceptable community-based intervention trial. I have the following observations and suggestions:

5a. Sampling method and randomization: As I understand, the authors used a purposive (or convenience) sampling method to include 14 clinics, and the clinics were distributed randomly into two groups - 7 intervention clinics and 7 control clinics. 

It is important to describe how the randomization was done.

5b. Is it possible to describe the average distance between the intervention and the control clinics? A suitable buffer zone may help in reducing any possible diffusion of information between the two groups. Please mention this in the limitations, if this cannot be maintained.

5c. IRB clearance and ethical procedures should be mentioned in the method section (although mentioned at the end of the manuscript, lines 262-264)

6. A CONSORT Flow Diagram (Figure 1) should be added to visualize the methods. 

7. Results - described adequately.

8. Discussion:

Strength and Limitations: Please mention the limitations of generalizability when a purposive/convenience sample method is used.

9. Conclusion: Minor - Please spell out SSA used the first time (line 248).
